# Health of Young Adults Experiencing Social Marginalization and Vulnerability: A Cross-National Longitudinal Study

**DOI:** 10.3390/ijerph20031711

**Published:** 2023-01-17

**Authors:** Jessica A. Heerde, Gabriel J. Merrin, Vi T. Le, John W. Toumbourou, Jennifer A. Bailey

**Affiliations:** 1Department of Paediatrics, The University of Melbourne, Parkville 3052, Australia; 2Department of Social Work, The University of Melbourne, Parkville 3010, Australia; 3Centre for Adolescent Health, Murdoch Children’s Research Institute, Parkville 3052, Australia; 4Department of Human Development and Family Science, Syracuse University, Syracuse, NY 13244, USA; 5Social Development Research Group, School of Social Work, University of Washington, Seattle, WA 98115, USA; 6Centre for Social and Early Emotional Development, School of Psychology, Deakin University, Burwood 3125, Australia

**Keywords:** homelessness, adverse childhood experiences, LGBT, justice system involvement, financial insecurity, young adults, marginalization, physical health, mental health, substance use, longitudinal

## Abstract

People who experience social marginalization and vulnerability have uniquely complex health needs and are at risk of poor health outcomes. Regression analyses using longitudinal data from a cross-national, population-based sample of young adults participating in the International Youth Development Study, tested associations between social marginalization and vulnerabilities and physical health, mental health, and substance use outcomes. Participants from Victoria, Australia, and Washington State in the US were surveyed at ages 25 (2014) and 29 years (2018; *N* = 1944; 46.7% male). A history of adverse childhood experiences (ACEs), LGBT identity, financial insecurity, and justice system involvement at age 25 predicted poor health outcomes at age 28, including lower perceived health status, risk for chronic illness, depression and anxiety symptoms, and diagnosed mental health/substance use disorders. Tests of model equivalence across states showed that a history of ACEs was more strongly related to health status and serious injury at age 28 and justice system involvement at age 25 was more strongly related to age 28 serious injury in Victoria than in Washington State. Findings strengthen the case for future population-based research identifying life-course interventions and state policies for reducing poor health and improving health equity among members of socially marginalized groups.

## 1. Introduction

Young adulthood is a critical period of development during which skills are built to achieve later health, social and economic wellbeing [1]. Yet, young adults often face significant challenges that may interrupt their development. For example, mental ill-health and substance use rates usually peak during this period [2]. Rates of homelessness and housing instability among young adults remain concerning and are associated with preventable morbidity [3]. Social marginalization and vulnerability at this developmentally critical period are likely to have significant consequences for later adult health and health inequities [4,5], presenting a strong imperative for understanding the health impacts of young adult social marginalization and vulnerability.

Social marginalization occurs when individuals or groups of people are excluded from mainstream society, for example because of poverty, structural disadvantage and discrimination. People experiencing marginalization often experience less access to services (including health care) and opportunities for social participation [6]. Marginalization is often a cause of vulnerability, described as increased risk for harm or poor outcomes and driven by socio-cultural, environmental, socio-economic and political factors [7]. Previous research, much of it cross-sectional in nature and conducted using retrospective data collected from small, selected samples [3,8], has documented the disproportionate burden of poor health among young adults who experience social marginalization and associated vulnerabilities.

Existing literature reviews suggest young people experiencing homelessness report poor physical and mental health [9], high rates of substance use [10], morbidities related to communicable and non-communicable diseases, and violence and victimization [11,12]. Other reviews have associated adverse childhood experiences (ACEs) with later disease burdens, including poor mental and physical health, problematic substance use [13,14,15], and homelessness [16]. Prior studies also suggest lesbian, gay, bisexual, and transgender (LGBT) individuals experience heightened health concerns, including elevated risk for substance use, poor mental health, reduced general health, and homelessness [17,18]. A recent review of the health of young people involved in the justice system found a high lifetime prevalence of various physical and mental health problems, such as substance use disorders, deliberate self-harm, and infectious diseases [19]. The various physical and mental health impacts of financial insecurity, such as depression and anxiety, substance abuse, violence and poor physical health, have also been reported [20].

The health of groups experiencing social marginalization and vulnerability continues to receive increasing international attention. Deepening our understanding of the health challenges among these groups is required to better inform efforts to address broader structural causes that impact health inequity and to inform the provision of health and support services. Longitudinal population-based studies offer a critical opportunity to provide a better understanding of the strength of the association between marginalization and vulnerability and later health outcomes, which are likely biased in studies analyzing retrospective data. Further, examining whether these associations are similar or different across international contexts will provide important information on whether effects are universal or subject to differences due to cultural and state policy variation. Cross-national comparisons of findings from prospective studies offer these important benefits [21,22], as well as the opportunity for within-study replication of results [23]. Studies such as these are rare, particularly focussing on health equity [24], despite their utility to inform policy and test theories of the influence of marginalization and vulnerability on health. The current study uses data from a large international cross-national general population sample: the International Youth Development Study (IYDS). The IYDS is an ongoing cross-national longitudinal study investigating the development of healthy and problem behaviors among participants in Washington State (USA) and Victoria (Australia). The two states were originally chosen for their different policy environments around substance use but are similar on several socio-demographics characteristics. Australia adopts a harm minimization approach to substance use, in contrast to the abstinence approach adopted in the United States [23]. Our prior IYDS studies found higher rates of young adult alcohol use in Victoria, Australia compared to Washington State, USA [25]. It is possible these state elevations in alcohol use increase inequalities.

In the current study, we examine whether social marginalization and vulnerability at age 25 are prospectively associated with risk for poor health outcomes at age 28 using data from the IYDS. Two research questions were examined: (1) To what extent do social marginalization and vulnerability predict poor health outcomes in young adulthood? and (2) Is the predictive nature of these associations similar in Washington State in the USA and Victoria, Australia?

## 2. Materials and Methods

### 2.1. Sample

Data in this study were collected from young adults participating in the IYDS. Participants were initially recruited as adolescents in state-representative secondary school samples in 2002 (age 12 years, Grade/Year 7) and have been followed to 2018–2019 (age 28 years).

The original IYDS sampling and recruitment methods have previously been described [23]. The study design was subjected to several processes in 2001 to ensure cross-national validity and reduce methodological problems identified in prior international comparison studies [22]. These processes included: matched sampling and recruitment strategies, matched surveys and survey administration procedures, and cognitive pretesting and piloting of the survey (including language review and cross-national item adaptation) [23]. Recruitment of state-representative samples was achieved using a two-stage cluster sampling approach in 2002: (1) public and private schools with Grades/Years 5, 7, and 9 (youngest, middle, and oldest cohorts, respectively) were randomly selected for recruitment using a probability proportionate to grade-level size sampling procedure [26]; and (2) one class at the appropriate grade level were randomly selected within each school [23]. Across all three cohorts, 7782 eligible students (3856 in Washington State) were approached to participate, of which 2885 in Washington State and 2884 in Victoria (74.8% and 73.5%, respectively) consented to and took part in the 2002 survey. At the time of recruitment, demographic and economic characteristics of population size and urbanicity, higher than national levels of educational participation, and low proportion of residents living in poverty, were similar in both states [23]. 

Data analyzed in this study were collected from the middle cohort (Grade/Year 7 in 2002) at ages 25 and 28 years (in 2014–2015 and 2018–2019, respectively); this was the only cohort followed longitudinally in both states. Retention rates across both states were at least 83% at these follow-up surveys. The analysis sample includes 1944 young adults ranging in age from 23.77 to 27.42 at the 2014–2015 survey (*M* = 25.2, SD = 0.48; 46.7% male). Participants self-reported their current educational status at the 2014–15 survey; 84.82% were not currently studying (5.81% studying part-time, 5.68% studying full-time).

### 2.2. Procedure

Approval for this study was obtained from the University of Melbourne Human Ethics in Research Committee in Australia and the University of Washington Human Subjects Institutional Review Board in Washington State. Written parental consent and participant assent were obtained at the study’s outset in 2002. Participant consent also was obtained for each of the young adult surveys. Self-report young adult surveys were completed online and took 50–60 min. Participants were reimbursed USD/AUD 40 for their time (at each survey).

### 2.3. Measures

The IYDS survey used self-report measures of young adult health and behavior adapted from the Communities That Care youth survey [27,28]. Participant demographic data were also collected. Survey measures were reviewed and adjusted to be developmentally appropriate as the study sample aged from adolescence into young adulthood. The IYDS survey and measures analyzed in this study have demonstrated longitudinal validity and reliability in the Victorian and Washington State samples [29,30].

### 2.4. Socially Marginalized and Vulnerable Groups

*Homelessness (Age 25)* was measured using two items: “In the past year, have you been homeless (i.e., not had a regular place to live)?” and “Which of the following best describes where you currently live?” In line with international definitions of homelessness [31], homelessness was scored as 1 (otherwise 0) if respondents replied yes to the first question or responded that they were, for example, currently ‘staying with friends temporarily’ or living in a ‘refuge/temporary accommodation,’ or ‘hotel/motel/caravan’. Prior studies demonstrate these items accurately reflect the forms of homelessness young adults may experience (e.g., being unsheltered, couch surfing, residing in temporary accommodation) [32].

*Adverse Childhood Experiences* (ACEs) were measured retrospectively at age 28 [33]. Ten items assessed 8 ACEs, including physical, sexual, and psychological abuse (perpetrated by a parent or adult); parental or adult: substance use or mental health problems; incarceration of a household member; and loss of a biological parent (for example, through death or divorce). A sum score indexing number of ACEs experienced was included in the analyses.

*Sexual identity (LGBT identity, Age 25).* Participants reported their sexual identity in response to the item, ‘Please choose the description that best fits how you think about yourself.’ Descriptions included: 100% heterosexual (straight); Mostly heterosexual (straight) but somewhat attracted to people of your own sex; Bisexual, that is, attracted to men and women equally; Mostly homosexual (gay or lesbian), but somewhat attracted to people of the opposite sex, and 100% homosexual. Response options were recoded to reflect ‘not LGBT’ (0, reference group) and ‘LGBT’ (1, comparison group).

*Financial insecurity (Age 25)* was measured using four items. “Which one of the following best describes your current financial situation?” measured participants’ *current financial situation*. Response options included ‘living comfortably’ (1), ‘doing alright’ (2), ‘just about getting by’ (3), ‘finding it quite difficult’ (4), and ‘finding it very difficult’ (5) and were recoded to reflect ‘No financial problems’ (1–2, 0 reference group) and ‘Yes, financial problems’ (3–5, 1 comparison group). *Employment status* was assessed using the item “Are you currently employed?” (reversed so that ‘Yes’ = 0 and ‘No’ = 1). “What is your usual, take-home, weekly income from all sources of support?” was used to measure *weekly income* (ranging from USD/AUD 0 to 2001+). Response options were recoded to reflect ‘middle-high income’ (0, top 75% of weekly income) and ‘low income’ (1, bottom 25% of weekly income). Receipt of *government benefits* was measured using the item: “What are your sources of income: Government allowance?” Response options were dichotomous, ‘yes’ (1, comparison group) or ‘no’ (0, reference group). Qualitative responses for receipt of government benefits were cross-examined by two authors (J.A.H & J.A.B) and recoded accordingly. Dichotomized scores for the four indicators of financial insecurity (i.e., financial status, employment status, weekly income, receipt of government benefits) were summed to create a total financial insecurity score ranging from 0 to 4.

*Justice system involvement (Age 25)* was assessed using three items: ‘How many times in the past 12 months have you been cautioned by police?’, ‘How many times in the past 12 months have you been charged by police?’ and ‘How many times in the past 12 months have you appeared in court for a criminal offense?’ Response options for each of these items were ‘Never’ (1), ‘1–2 times’ (2), ‘3–5 times’ (3), ‘6–9 times’ (4), and ‘10 or more times’ (5). Items were combined to index ‘no justice system involvement’ (0, reference group) and ‘justice system involvement’ (1, comparison group).

### 2.5. Health Outcomes (Age 28)

*Perceived health status* was measured using the item, ‘In general, would you say your health is?’ Response options included: ‘Excellent’ (1),’ Very good’ (2), ‘Good’ (3), ‘Fair’ (4) and ‘Poor’ (5). 

*Chronic illness* was assessed using the item, ‘Have you ever had any of the following illnesses or conditions?’: diabetes, high cholesterol requiring medication, high blood pressure, asthma, glaucoma, hepatitis or liver disease, seizures, a thyroid condition, ulcer, or none of the above. Dichotomized scores for each response option were summed to produce a cumulative count of illnesses or conditions, with scores ranging from 0 to 6.

*Serious injury* was measured using the item, ‘In the past 12 months, have you had a serious physical injury that required medical attention (e.g., bandaging, stitches, loss of a tooth, broken bones, or amputation) related to: sports, injury sustained in the workplace or while working, or any other accident?’ Response options were dichotomous, ‘no’ (0, reference group) and ‘yes’ (1, comparison group). As a follow-up, participants were asked if they had ‘experienced any *disability or ongoing medical problems* because of the injury (or injuries)?’. Response options were dichotomous, ‘no’ (0, reference group) and ‘yes’ (1, comparison group).

*Depression and anxiety symptoms* were measured using the Kessler Psychological Distress Scale (K10) [34]. Items were answered on a 5-point scale of ‘None of the time’ (1), ‘A little of the time’ (2), ‘Some of the time’ (3), ‘Most of the time’ (4) and ‘All of the time’ (5). Scores across all items were summed then recoded using standard cut points: ‘<20 no symptoms’ (0, reference group),’ 20–24 mild symptoms’ (1), ‘25–29 moderate symptoms’ (2) and ‘≥30 severe symptoms’ (3) [34].

*Diagnosed mood, anxiety, and substance use disorders* were assessed by asking respondents whether they had ever been diagnosed by a mental health clinician with a mood disorder, or an anxiety disorder, or a substance abuse disorder. Response options were dichotomous, ‘no to all’ (0, reference group) and ‘yes to any’ (1, comparison group).

*Demographic factors.* Participants reported their *age* (date of birth), *sex* (female, 0 or male, 1) and the *state* in which they lived (Victoria, 0 or Washington State, 1) at each survey. *Family socioeconomic status* (SES) was created using measures of parent (mother and father) self-reported level of annual family income (ranging from <USD/AUD10,000 to =>200,000) and the highest level of education completed (for example, less than secondary school, completed secondary school, and completed post-secondary school). These two measures (levels of annual family income and level of education completed) were obtained in phone interviews conducted with parents around the time their adolescent child took part in the 2002 (baseline) survey. *Race and ethnicity* are conceptualized differently in Australia and the USA. Australian participants self-reported their race/ethnicity in response to the item ‘What do you consider yourself to be?’ Response options included ‘African,’ ‘Aboriginal or Torres Strait Islander,’ ‘Spanish/Hispanic/Latino,’ ‘Asian,’ ‘Pacific Islander,’ ‘Australian,’ and ‘other.’ US participants were asked, ‘What best describes your racial background.’ Response options included ‘White,’ ‘Black or African American,’ ‘American Indian/Native American or Alaskan Native,’ ‘Asian,’ ‘Native Hawaiian or other Pacific Islander,’ and ‘Other.’ Given the variation in the conceptualization of race/ethnicity across the two countries, a dichotomous variable of minoritized (nonwhite [1, comparison group]) versus non-minoritized (white [0, referent group]) participants is used for statistical models where original categories are used for descriptives.

### 2.6. Statistical Analysis

All analyses were conducted using Mplus 8.7 [35]. We fit a series of unadjusted and adjusted regression models (continuous and logistic) to examine the associations between social marginalization and vulnerabilities (age 25) and physical health, mental health, and substance use outcomes (age 28). The unadjusted model assessed each of the independent variables (i.e., age, sex, race, country, SES, homelessness, ACEs, LGBT, financial insecurity, justice system involvement) separately on each of the physical health (i.e., poor perceived health status, chronic illness, serious injury, disability or ongoing medical problems resulting from serious injury), mental health (depression and anxiety symptoms, diagnosed mood disorder, diagnosed anxiety disorder), and substance use (i.e., diagnosed substance use disorder) outcomes. Following, adjusted models that controlled for all variables were fitted. As a final step, to examine the predictive nature of these associations cross-nationally, we fit a series of adjusted models for each state using groups analyses. Differences in the magnitude of these effects between the two states were examined using Wald tests.

Logistic link functions with a robust maximum likelihood estimator were used to examine dichotomous health outcomes. For categorical health outcomes, a weighted least squares mean and variance adjusted estimator was used to fit a continuous version of the variable while accounting for its categorical nature. Full information maximum likelihood was used to minimize potential bias due to missing data in models using the robust Maximum Likelihood Estimator, with pairwise deletion used in models using the weighted least squares means and variance adjusted estimator [35]. The percentage of missing data on the analyzed variables ranged from 0.6 to 43%.

## 3. Results

Descriptive statistics are presented in Table 1 for the full sample and for participants in each state (Victoria and Washington State), including an overview of the race and ethnic identities of the study sample [35].

### 3.1. Physical Health Outcomes

Table 2 presents the adjusted results for associations between age 25 social marginalization and vulnerability and age 28 physical health outcomes (poor perceived health status, chronic illness, serious injury, and disability or ongoing medical problems resulting from serious injury). Older age, male sex, and race (nonwhite) were not significantly associated with any physical health outcomes. Higher family SES in adolescence was significantly associated with higher perceived health status and lower odds of serious injury in young adulthood. Living in Washington State was associated with lower odds of chronic illness compared to living in Victoria.

Having experienced a greater number of ACEs was significantly associated with lower perceived health status and higher odds of chronic illness and disability or ongoing medical problems resulting from serious injury, but not serious injury at age 28. LGBT status was significantly associated with lower perceived health status compared to heterosexual participants, but not chronic illness, serious injury, or disability or ongoing medical problems resulting from serious injury. Age 25 financial insecurity was associated with lower perceived health status and chronic illness but not serious injury or disability or ongoing medical problems resulting from serious injury at age 28.

Homelessness at age 25 was significantly associated with higher odds of serious injury at age 28, and justice system involvement at age 25 was significantly associated with higher odds of later serious injury and disability or ongoing medical problems resulting from serious injury in the unadjusted models. However, these associations did not reach statistical significance in the fully adjusted models (see Appendix A).

### 3.2. Mental Health and Substance Use Outcomes

Table 3 presents the adjusted results for associations between age 25 social marginalization and vulnerability and age 28 mental health (depression and anxiety symptoms, diagnosed mood disorder, diagnosed anxiety disorder) and substance use (diagnosed substance use disorder) outcomes. Older age was not significantly associated with any age 28 mental health and substance use outcomes. Males (compared to females) and nonwhite individuals (compared to white) reported lower rates of depression and anxiety symptoms and lower odds of both diagnosed mood and anxiety disorders, respectively. Higher family SES in adolescence was significantly associated with a higher likelihood of a diagnosed anxiety disorder. Living in Washington State was associated with higher odds of a diagnosed substance use disorder compared to living in Victoria.

A greater number of ACEs and financial insecurity at age 25 were associated with higher rates of age 28 depression and anxiety symptoms and higher odds of age 28 diagnosed mood, anxiety, and substance use disorders, respectively. Identifying as LGBT was associated with higher rates of self-reported depression and anxiety symptoms and higher odds of diagnosed mood and anxiety disorders, but not diagnosed substance use disorder at age 28. Justice system involvement at age 25 was associated with higher odds of a diagnosed substance use disorder at age 28 but not self-reported depression and anxiety symptoms, or a diagnosed mood or anxiety disorder.

Homelessness at age 25 was significantly associated with higher rates of self-reported depression and anxiety symptoms at age 28 and higher odds of a diagnosed anxiety disorder and substance use disorder at age 28 in unadjusted analyses, but these associations did not reach statistical significance in the fully adjusted models (see Appendix A). Similarly, justice system involvement at age 25 was significantly associated with higher rates of self-reported depression and anxiety symptoms and higher odds of a diagnosed substance use disorder at age 28 in the unadjusted models; these associations were not statistically significant in the fully adjusted models.

### 3.3. Groups Analysis

Tests of model equivalence across states showed potential state differences in two models: perceived health status and serious injury. A greater number of ACEs was significantly associated with serious injury in Victoria but not Washington State. Specifically, the effect of ACEs on higher levels of perceived health status at age 28 was larger in Victoria compared to Washington State (*p* = 0.015). Higher family SES in adolescence, a greater number of ACEs, and age 25 justice involvement were significantly associated with serious injury for Victoria but not Washington State. Specifically, the size of the effect of family SES in adolescence on lower levels of serious injury at age 28 was larger for Victoria compared to Washington state (*p* = 0.045). In addition, the positive associations between ACEs (*p* = 0.020) and age 25 justice involvement (*p* = 0.032) on age 28 serious injury were larger in Victoria.

## 4. Discussion

Social marginalization and vulnerabilities remain major contributors to poor health and health inequity. Despite the international recognition that groups experiencing social vulnerabilities are subject to health inequity and have poor health outcomes, health and social policies in many countries fall well short of need [24]. Research that generates evidence to support a stronger focus on health impacts and identifying intervention entry points that reduces these vulnerabilities and inequities is needed. The current study is one of few to examine the health impacts of young adult social marginalization and vulnerability using longitudinal data collected from a cross-national population-based sample. Models showed ACEs, LGBT identity, financial insecurity, and involvement with the justice system were longitudinally associated with poor health outcomes, including lower perceived health status, risk for chronic illness, depression and anxiety symptoms, and diagnosed mental health/substance use disorders. Family socioeconomic status in adolescence also showed health impacts into young adulthood. Cross-national differences in some associations were identifiable. The findings suggest experiencing social marginalization and vulnerability heightens the risk for poor health outcomes in young adulthood that are likely to underpin health across the later life course.

The study findings demonstrate the feasibility of using longitudinal, population-level data to examine the extent to which social marginalization and vulnerability are related to various health outcomes. In this study, a history of ACEs was acutely related to the majority of health outcomes examined, both self-reported physical health and illnesses and conditions diagnosed by a mental health clinician. ACEs, traumatic events that occur in childhood, have been linked to a range of chronic health problems, mental ill-health, and problematic substance use [27]. The findings of the current study reflect the body of existing research that has reported these associations. Indeed, the literature reporting on childhood experiences of individuals with membership in the socially vulnerable and marginalized groups examined in this study (homeless young adults, LGBT identity, people involved with the justice system) suggests that ACEs (such as not having received adequate family care and protection, family breakdown, childhood abuse) are an established risk factor for marginalization and poor health among members of these groups [16,36,37]. Our findings emphasize the importance of prevention and intervention efforts targeted toward reducing the occurrence of ACEs at the population level. Equally, our findings illustrate the need for the analysis of large-scale longitudinal population-level data (as discussed below) to examine the interplay between ACEs, membership in socially vulnerable groups, and risk for many health conditions to: (a) better inform prevention programming at the earliest possible point in the life course and (b) assess the longer-term health impacts of ACEs on health across the life course.

Prior studies have noted the significant health impacts of experiencing homelessness and of justice system involvement [8,15]. At the univariable level, both young adult homelessness and involvement with the justice system were associated with most health outcomes investigated; however, only justice system involvement and risk for a diagnosed substance use disorder showed a statistically significant multivariable association. The low prevalence of both young adult homelessness and justice system involvement in the current sample may be one reason for these findings. Equally, as discussed in relation to ACEs, it is plausible that there are overlapping risks across the socially marginalized groups examined in the current study. Participants in this study are likely members of multiple marginalized groups (e.g., report a history of ACEs and homelessness and justice system involvement). As such, it is expected these young adults experience layers of risk related to their membership in each of these groups [38] that influences risk for poor health outcomes. Subsample sizes were insufficient to investigate intersectionality across groups in the current study. Future studies should employ an intersectional lens that considers membership in overlapping socially marginalized groups to understand how membership in one group may contribute to membership in other groups and consequently heighten the risk for poor health outcomes.

This study is distinct from prior studies of health among socially marginalized groups in that we have examined cross-national differences in the predictive nature of social marginalization and vulnerability and poor health outcomes in young adulthood. Given different cultural contexts, social safety nets, and healthcare approaches (i.e., nationalized in Australia, privatized in the USA), it was possible that the impacts of marginalization and social vulnerability might differ in the two countries. In fact, the experience of ACEs was more strongly related to young adult outcomes in Australia compared to the US. However, overall results suggested a similar cross-national pattern of association between social marginalization and vulnerability indicators and many of the outcomes examined. This underscores a commonality, at least in developed, Western countries, in the health and behavioral impacts of membership in vulnerable and minoritized groups and suggests that similar prevention priorities and strategies may be effective in multiple national contexts.

Pathways into social marginalization and vulnerability are undoubtedly transactional and multi-faceted, reflecting the critical importance of understanding and addressing multiple risks through evidence-based practice. Our analyses focused exclusively on testing the theorized associations between social marginalization and vulnerability and young adult health outcomes. Building on these findings, a next logical step is the mapping of longitudinal pathways to marginalization and vulnerability (e.g., homelessness, justice system involvement) and testing the role of social-ecological risk and protective factors that influence risk for marginalization and vulnerability and how these factors increase or moderate risk for later health outcomes. In efforts to prevent or reduce marginalization and vulnerability and later health outcomes, risk and protective factors could then be identified and emphasized as influences that may reduce the likelihood of marginalization and/or poor health. As one example, a large body of research has identified disparities in substance use across the life course among individuals who experience social marginalization and vulnerability, such as people experiencing homelessness [39], those who have experienced ACEs [15,40], people who have had contact with the justice system [41], those who experienced childhood poverty [42] or people who identify as LGBT [43]. Less work has examined trajectories of substance use within these groups or predictors of persistence and desistence across adulthood. Yet, a greater understanding of longitudinal patterns of substance use within these groups and the factors that promote desistence from substance use among members of marginalized groups would inform prevention and treatment interventions aimed at promoting positive functioning and recovery from substance misuse.

One of the limitations of existing research, particularly with people experiencing homelessness [3] and those who have had contact with the justice system [8], has been a reliance on cross-sectional retrospective data collected from small convenient samples without a population-sample comparison group. This results from challenges associated with obtaining comprehensive population data from those that experience high levels of marginalization and vulnerability and are often transient and difficult to engage and retain in large, prospective cohort studies [3]. In this context, new strategies of data linkage studies [44,45] and multi-cohort life-course approaches [46] present exciting opportunities to enrich the range and type of health, social and behavioral data from which inequalities, health status, and the impact of health care can be studied. Both approaches permit the bringing together of large longitudinal population-level data and tackle many of the limitations of prior research, including the low prevalence of social marginalization and vulnerability seen in prior population-based cohort studies, e.g., [30,46,47]. These strategies are key methods for documenting the health of vulnerable and marginalized groups, as well as identifying population-level drivers of marginalization and vulnerability, socio-ecological risk and protective processes, and the developmental timing of these drivers and processes. Such information is critical for identifying targets for evidence-based population level interventions designed to reduce health inequities.

## 5. Strengths and Limitations

This study has several strengths. First, it analyses data collected from population-based samples, recruited to be state-representative in 2002. The study examined two cross-state samples that used identical methods in recruitment, surveying, and longitudinal follow-up [23]. The study has achieved excellent response rates in adolescence and has maintained high retention rates into young adulthood. The IYDS survey demonstrates longitudinal reliability and validity in both the Washington State and Victorian Samples [29,30]. Consequently, the study capitalizes on a unique opportunity to examine associations between social marginalization and vulnerability and later health outcomes, cross-nationally.

Several study limitations are acknowledged. It is noted that rates of young adult homelessness and involvement with the justice system are likely subject to underestimation. We did not conduct survey visits at correctional facilities, and other community-based services at the young adult follow-up points. Although study retention rates were high, it is possible that those experiencing homelessness or those having been involved in the justice system were less likely to participate in the study over time. These are both high-risk groups for attrition. Previous studies using the young adult sample analyzed here showed that although those experiencing homelessness were significantly less likely to be retained in the study, the overall retention of this group remained high (approx. 82%), and the difference in retention between young adults reporting and not reporting homelessness was small [48]. Owing to the low prevalence of homelessness and justice system involvement, our analyses may have been underpowered to detect associations between these and later health outcomes. Small sample sizes within some of the groups experiencing social marginalization analyzed here limited our capacity to examine health outcomes among members of multiple marginalized groups. Further, testing the role of socio-ecological risk and protective factors that influence risk for marginalization and vulnerability and those that increase or moderate risk for later health outcomes were beyond the scope of the current study. For example, we were unable to look at the potential moderating effect of access to health insurance. Further research which identifies these factors and the effect they have on marginalization/health is required. This study analyzed self-report data; this is considered reliable in studies of young adults [49]. Last, our findings are generalizable only to the state and cohort samples analyzed.

## 6. Conclusions

Social marginalization and vulnerabilities are major international contributors to poor health and health inequity. The current study is unique in its use of data from a cross-nationally matched population-based sample of young adults to examine associations between social marginalization and vulnerability and later health. Adverse childhood experiences and LGBT identity were associated with various physical health outcomes and diagnosed mental health conditions. The health impacts of experiencing homelessness, having contact with the justice system, and multiple marginalities, require future investigation. Findings derived from these investigations are vital for establishing the evidence base for identifying intervention points that reduce poor health and improve health equity among members of socially vulnerable and marginalized groups.

## Figures and Tables

**Table 1 ijerph-20-01711-t001:** Characteristics of the study population by nation.

	Combined Sample	Victorian Sample	Washington State Sample
Demographics (baseline)			
Age, mean (SD)	25.2 (0.48)	25.1 (0.46)	25.3 (0.46)
Sex, n (%)			
Female	907 (53.1%)	457 (52.8%)	450 (53.5%)
Male	800 (46.9%)	409 (47.2%)	391 (46.5%)
Family socioeconomic status, mean (SD)	2.01 (0.43)	1.96 (0.48)	2.07 (0.37)
Race and ethnicity—Victoria			
Aboriginal or Torres Strait Islander		10 (1.0%)	
African		7 (0.7%)	
Asian		51 (5.3%)	
Australian		880 (90.6%)	
Spanish/Hispanic/Latinx		4 (0.4%)	
Pacific islander		9 (0.9%)	
Other		10 (1.0%)	
Race and ethnicity—Washington State			
Asian			36 (3.7%)
American Indian/Native American or Alaskan Native			46 (4.9%)
Black or African American			32 (3.4%)
Latinx			95 (10.2%)
Multiracial			66 (7.1%)
Native Hawaiian or other Pacific Islander			15 (1.6%)
White			643 (68.9%)
Social marginalization and vulnerability (Age 25)			
Homelessness			
Yes	89 (5.2%)	36 (4.2%)	53 (6.3%)
No	1608 (94.8%)	823 (95.8%)	785 (93.7%)
ACEs, mean (SD)	1.46 (2.03)	1.31 (1.87)	1.61 (2.17)
Sexual identity, n (%)			
LGBT	329 (19.4%)	163 (19.0%)	166 (20.0%)
Not LGBT	1363 (80.6%)	697 (81.0%)	666 (80.0%)
Financial insecurity, mean (SD)	0.83 (1.07)	0.81 (1.12)	0.84 (1.02)
Justice system involvement, n (%)			
Yes	172 (15.7%)	103 (12.0%)	69 (29.6%)
No	921 (84.3%)	757 (88.0%)	164 (70.4%)
Health outcomes (Age 28)			
Perceived health status, mean (SD)	2.61 (0.94)	2.64 (0.96)	2.57 (0.92)
Chronic illness, n (%)			
Yes	458 (29.0%)	253 (31.9%)	205 (26.0%)
No	1124 (71.0%)	541 (68.1%)	583 (74.0%)
Serious injury, n (%)			
Yes	86 (5.4%)	36 (4.5%)	50 (6.3%)
No	1502 (94.6%)	761 (95.5%)	741 (93.7%)
Disability or ongoing medical problems resulting from serious injury, n (%)			
Yes	55 (3.5%)	21 (2.6%)	34 (4.3%)
No	1529 (96.5%)	775 (97.4%)	754 (95.7%)
Self-reported depression and anxiety symptoms, n (%)			
None	875 (55.4%)	425 (53.4%)	450 (57.4%)
Mild	301 (19.1%)	163 (20.5%)	138 (17.6%)
Moderate	193 (12.2%)	96 (12.1%)	97 (12.4%)
Severe	211 (13.4%)	112 (14.1%)	99 (12.6%)
Diagnosed mood disorder, n (%)			
Yes	319 (20.2%)	165 (20.7%)	154 (19.7%)
No	1260 (79.8%)	631 (79.3%)	629 (80.3%)
Diagnosed anxiety disorder, n (%)			
Yes	325 (20.6%)	160 (20.1%)	165 (21.1%)
No	1254 (79.4%)	636 (79.9%)	618 (78.9%)
Diagnosed substance use disorders, n (%)			
Yes	45 (2.8%)	8 (1.0%)	37 (4.7%)
No	1534 (97.2%)	788 (99.0%)	746 (95.3%)

Note. Family socioeconomic status (SES) in adolescence ranged from 1–3 (higher scores indicating higher socioeconomic status). Multiracial includes people who identified with multiple racial and ethnic identities (e.g., Native-White, Latinx-White, Latinx-Native). Adverse Childhood Experiences (ACEs) ranged from 0–10 (higher scores indicating more adverse experiences). Financial insecurity ranged from 0–4 (higher cores indicating more financially insecure). Perceived health status ranged from 1–5 (higher scores indicated poorer perception of general health). Depression and anxiety symptomology were measured using the Kessler Psychological Distress Scale, and the number of total symptoms were classified as: ‘None: <20 no symptoms’ (0), ‘Mild: 20–24, mild mental disorder’ (1), ‘Moderate: 25–29, moderate mental disorder’ (2) and ‘Severe: ≥30, severe mental disorder’.

**Table 2 ijerph-20-01711-t002:** Adjusted associations between social marginalization and vulnerability (Age 25) with general and physical health outcomes at age 28.

	Poor Perceived Health Status		Chronic Illness		Serious Injury		Disability or Ongoing Medical Problems Resulting from Serious Injury	
	b (SE)	*p*	OR (SE)	*p*	OR (SE)	*p*	OR (SE)	*p*
Social marginalization and vulnerability								
Homelessness	−0.10 (0.11)	0.358	0.95 (0.27)	0.853	1.50 (0.67)	0.367	0.70 (0.52)	0.632
ACEs	**0.05** (0.01)	**<0.001**	**1.14** (0.03)	**<0.001**	1.10 (0.06)	0.083	**1.12** (0.06)	**0.049**
LGBT	**0.26** (0.07)	**<0.001**	1.14 (0.17)	0.373	0.75 (0.24)	0.365	0.63 (0.27)	0.281
Financial insecurity	**0.14** (0.03)	**<0.001**	**1.13** (0.06)	**0.033**	1.15 (0.11)	0.164	1.15 (0.17)	0.338
Justice system involvement	0.06 (0.10)	0.515	1.41 (0.29)	0.097	1.89 (0.69)	0.082	1.82 (0.84)	0.196
Demographic factors								
Age	0.09 (0.06)	0.167	1.15 (0.15)	0.298	1.02 (0.27)	0.928	1.22 (0.30)	0.426
Male	−0.02 (0.06)	0.771	0.89 (0.11)	0.345	1.42 (0.34)	0.14	1.25 (0.35)	0.434
Nonwhite	−0.15 (0.08)	0.072	0.75 (0.13)	0.094	1.02 (0.34)	0.946	0.83 (0.35)	0.653
Washington State	−0.08 (0.06)	0.155	**0.68** (0.08)	**0.002**	1.34 (0.34)	0.242	1.47 (0.47)	0.232
Family SES ^	**−0.31** (0.06)	**<0.001**	0.90 (0.12)	0.428	**0.55** (0.13)	**0.012**	1.17 (0.39)	0.627

Note. Models adjusted for all demographics and other types of social marginalization and vulnerability. Age was centered at the mean. Bolded effects are significant at *p* < 0.05. SES: socioeconomic status; b = unstandardized parameter estimate; SE: standard error; OR: odds ratio. ^ measured in adolescence.

**Table 3 ijerph-20-01711-t003:** Adjusted associations between age 25 social marginalization and vulnerability with mental and substance use outcomes at age 28.

	Self-Reported Depression and Anxiety Symptoms		Diagnosed Mood Disorder		Diagnosed Anxiety Disorder		Diagnosed Substance Use Disorder	
	b (SE)	*p*	OR (SE)	*p*	OR (SE)	*p*	OR (SE)	*p*
Social marginalization and vulnerability								
Homelessness	0.10 (0.12)	0.429	0.79 (0.25)	0.461	1.16 (0.36)	0.63	1.39 (0.72)	0.525
ACEs	**0.11** (0.01)	**<0.001**	**1.32** (0.04)	**<0.001**	**1.24** (0.04)	**<0.001**	**1.26** (0.08)	**<0.001**
LGBT	**0.37** (0.07)	**<0.001**	**2.14** (0.33)	**<0.001**	**1.79** (0.28)	**<0.001**	1.67 (0.64)	0.186
Financial insecurity	**0.16** (0.03)	**<0.001**	**1.37** (0.09)	**<0.001**	**1.26** (0.08)	**<0.001**	**1.57** (0.19)	**<0.001**
Justice system involvement	0.17 (0.10)	0.087	0.61 (0.17)	0.075	1.17 (0.31)	0.538	**3.75** (1.88)	**0.008**
Demographic factors								
Age	0.06 (0.06)	0.372	0.97 (0.16)	0.826	0.77 (0.12)	0.093	0.58 (0.25)	0.211
Male	**−0.13** (0.06)	**0.026**	**0.60** (0.09)	**<0.001**	**0.37** (0.06)	**<0.001**	1.07 (0.39)	0.86
Nonwhite	**−0.18** (0.08)	**0.028**	**0.52** (0.12)	**0.003**	**0.59** (0.13)	**0.013**	0.65 (0.32)	0.377
Washington State	**−0.13** (0.06)	**0.024**	0.92 (0.14)	0.573	0.99 (0.14)	0.953	**4.21** (1.80)	**0.001**
Family SES ^	−0.06 (0.07)	0.346	1.30 (0.21)	0.105	**1.48** (0.23)	**0.012**	1.14 (0.51)	0.778

Note. Models adjusted for demographics and other types of social marginalization and vulnerability. Age was centered at the mean. Bolded effects are significant at *p* < 0.05. SES: socioeconomic status; b = unstandardized regression coefficient; SE: standard error; OR: odds ratio. ^ measured in adolescence.

## Data Availability

Requests regarding the data presented in this study should be directed to the corresponding author.

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
