# Peer review of "Health of Young Adults Experiencing Social Marginalization and Vulnerability: A Cross-National Longitudinal Study"

_ijerph, 2023, doi:10.3390/ijerph20031711_

Round 1

Reviewer 1 Report

The paper is very interesting, but in its current version it needs some modifications.I suggest the following:

a) Justify why the research was carried out in these two contexts

b) Describe the social conditions of both study contexts, especially thinking of international readers who are not familiar with the study contexts

c)The authors point out that the sampling and recruitment methods have been described in another of their papers, but if necessary it is explained in this one at least briefly.

d) In the section “2.4. Social vulnerability and marginalization” it is necessary to conceptualize what is vulnerability and marginalization. In this way, the way in which they measured these concepts can be better understood.

e) In the "Statistical Analysis" section, put the results of the fit tests or reference them in the table where they are. In the same way, it is necessary to formalize the statistical models used

f) In case of having the educational levels of the participants, add them in Table 1

h) The strengths do not need to be included, just leave the limitations. Put this section after the conclusions

i) General conclusions. These are not deep at all, it is important to add the social and public policy implications of the study

Author Response

Justify why the research was carried out in these two contexts.

To address the feedback regarding the two states chosen for this study, we have included additional text int the Introduction briefly summarising why the two states of Washington and Victoria were selected for comparison in the Study (lines 86-91).

Describe the social conditions of both study contexts, especially thinking of international readers who are not familiar with the study contexts.

We are in full agreement that describing the social conditions in both study contexts are important parts of discussions on reducing social marginalization and vulnerabilities. An in-depth discussion of these areas were beyond the scope of the current study. Congruent with our aim to explore the predictive nature of associations in the two states and the finding that there were few cross-national differences, we have briefly discussed cultural and health care approaches in the study sites in our Discussion (lines 410-422).

The authors point out that the sampling and recruitment methods have been described in another of their papers, but if necessary it is explained in this one at least briefly.

 To clarify, we have briefly summarised the sampling and recruitment methods in sections 2.1 and 2.2. of our manuscript. We have cited a previous published methods paper which discusses study sampling and recruitment, for readers who would like more detailed information.

In the section “2.4. Social vulnerability and marginalization” it is necessary to conceptualize what is vulnerability and marginalization. In this way, the way in which they measured these concepts can be better understood.

In response to this suggestion, and that of Reviewer 2, we have included text describing social marginalisation and vulnerability in the Introduction (lines 47-52). We have revised the subheading for section 2.4 to ‘Socially marginalized and vulnerable groups’ to align with the measures described in this section.

In the "Statistical Analysis" section, put the results of the fit tests or reference them in the table where they are. In the same way, it is necessary to formalize the statistical models used.

We thank the reviewer for this feedback. All models were fully saturated and thus have perfect fit, as such, there are no fit indices to present. The only formal comparisons that were made were in the groups analysis models in which we labeled all variables and set them to equality across countries and then used a Wald test to examine whether there were any country-level differences (akin to an omnibus F test). These differences are presented in the Results sections (refer to section 3.3).

In case of having the educational levels of the participants, add them in Table 1.

 Table 1 presents the sample characteristics for the variables analysed in the study. We have included a sentence in section ‘2.1 Sample’, describing the current educational status of participants (lines 132-133).

The strengths do not need to be included, just leave the limitations. Put this section after the conclusions.

Thank you for these suggestions. We have reviewed other papers recently published in the IJERPH, some of which briefly discuss study strengths. As such, we have retained our brief discussion of strengths in our manuscript.

 We have reviewed the journal guidelines for manuscript preparation. These guidelines state research manuscripts should include the sections: Introduction, Materials and Methods, Results, Discussion, Conclusions. Therefore, we have retained the Conclusions section as the final section of our manuscript.

 General conclusions. These are not deep at all, it is important to add the social and public policy implications of the study.

To clarify, section ‘6. Conclusions’ is a general closing paragraph summarising the main premise, results and findings of our study. We have comprehensively discussed social and public policy and research implications in section ‘4. Discussion’.

Reviewer 2 Report

This is an interesting article with a sound statistical technique. However, I think there are some points that can improve it:

a) Why did you choose these two regions/cities for this study?

b) You include ethnicity and sexual orientation, but not discrimination experiences in your analysis. Why?

c) It would be important to define what you understand about marginalization and vulnerability in the introduction. 

Author Response

Why did you choose these two regions/cities for this study?

To address the feedback regarding the two states chosen for this study, we have included additional text int the Introduction briefly summarising why the two states of Washington and Victoria were selected for comparison in the Study (lines 86-91).

You include ethnicity and sexual orientation, but not discrimination experiences in your analysis. Why?

We agree that it would be valuable to investigate discrimination experiences in our analyses; unfortunately, measures of discrimination were not available in the current study.

It would be important to define what you understand about marginalization and vulnerability in the introduction.

We thank the reviewer for this suggestion. Accordingly, we have included additional text in the Introduction contextualizing social marginalization and vulnerability (lines 47-52).

Round 2

Reviewer 1 Report

The authors made all corrections. Congratulations the paper is stronger now